# Sterile Insect Technique (SIT) against *Aedes* Species Mosquitoes: A Roadmap and Good Practice Framework for Designing, Implementing and Evaluating Pilot Field Trials

**DOI:** 10.3390/insects12030191

**Published:** 2021-02-24

**Authors:** Clélia F. Oliva, Mark Q. Benedict, C Matilda Collins, Thierry Baldet, Romeo Bellini, Hervé Bossin, Jérémy Bouyer, Vincent Corbel, Luca Facchinelli, Florence Fouque, Martin Geier, Antonios Michaelakis, David Roiz, Frédéric Simard, Carlos Tur, Louis-Clément Gouagna

**Affiliations:** 1Centre Technique Interprofessionnel des Fruits et Légumes (CTIFL), Centre Opérationnel de Balandran, 751 Chemin de Balandran, 30127 Bellegarde, France; clelia.oliva@ctifl.fr; 2Collectif TIS (Technique de l’Insecte Stérile), 751 Chemin de Balandran, 30127 Bellegarde, France; 3Mosquito Hunters, 620 Peachtree St, Atlanta, GA 30308, USA; mqbenedict@yahoo.com; 4Centre for Environmental Policy, Imperial College London, London SW7 1NE, UK; t.collins@imperial.ac.uk; 5ASTRE (Animal, Santé, Territoires, Risques, Ecosystèmes), Cirad, Univ Montpellier, 34398 Montpellier, France; thierry.baldet@cirad.fr (T.B.); bouyer@cirad.fr (J.B.); 6Centro Agricoltura Ambiente “Giorgio Nicoli”, S.r.l. Via Sant’Agata, 835, 40014 Crevalcore, Italy; rbellini@caa.it; 7Institut Louis Malardé, Papeete, 98713 Tahiti, French Polynesia; hbossin@ilm.pf; 8Insect Pest Control Laboratory, Joint FAO/IAEA Programme of Nuclear Techniques in Food and Agriculture, IAEA Vienna, Wagramer Strasse 5, 1400 Vienna, Austria; 9UMR MIVEGEC (Maladies Infectieuses et Vecteurs: Écologie, Génétique, Évolution et Contrôle), IRD-CNRS-Univ. Montpellier, 34394 Montpellier, France; vincent.corbel@ird.fr (V.C.); david.roiz@ird.fr (D.R.); frederic.simard@ird.fr (F.S.); 10Liverpool School of Tropical Medicine, Pembroke Place, Liverpool L3 5QA, UK; luca.facchinelli@lstmed.ac.uk; 11TDR (Special Programme for Research and Training in Tropical Diseases), WHO, 20 Avenue Appia, 1121 Geneva, Switzerland; fouquef@who.int; 12Biogents AG, Weissenburgstr. 22, 93055 Regensburg, Germany; martin.geier@biogents.com; 13Benaki Phytopathological Institute. 8, S. Delta str., Kifissia, 14561 Athens, Greece; a.michaelakis@bpi.gr; 14Grupo Tragsa–KM. 4,5 Bajo, A28476208-EMPRE, Moncada, 46113 Valencia, Spain; ctur@tragsa.es

**Keywords:** mosquito control, SIT, integrated vector management, stakeholder engagement, pilot trial

## Abstract

**Simple Summary:**

The sterile insect technique is familiar to agricultural pest management and is now being increasingly applied to mosquitoes as part of integrated vector management programs. This review leans on a growing literature and on the experience of its many authors to describe the key steps, and the challenges to be surmounted, in order to design and execute successful pilot studies in many environments. We emphasize integrating stakeholder mapping and engagement at all levels. Included are introductory descriptions of the key elements to (1) ensure wide stakeholder support through transparent communication plans and the identification of regulatory and financial frameworks; (2) select suitable field sites; (3) build a sound, and locally-adapted, integrated vector management strategy; (4) access the technical advancements to ensure high-quality releases; and (5) reliably assess the impacts and benefits of the field trial.

**Abstract:**

*Aedes albopictus* and *Aedes aegypti* are invasive mosquito species that impose a substantial risk to human health. To control the abundance and spread of these arboviral pathogen vectors, the sterile insect technique (SIT) is emerging as a powerful complement to most commonly-used approaches, in part, because this technique is ecologically benign, specific, and non-persistent in the environment if releases are stopped. Because SIT and other similar vector control strategies are becoming of increasing interest to many countries, we offer here a pragmatic and accessible ‘roadmap’ for the pre-pilot and pilot phases to guide any interested party. This will support stakeholders, non-specialist scientists, implementers, and decision-makers. Applying these concepts will ensure, given adequate resources, a sound basis for local field trialing and for developing experience with the technique in readiness for potential operational deployment. This synthesis is based on the available literature, in addition to the experience and current knowledge of the expert contributing authors in this field. We describe a typical path to successful pilot testing, with the four concurrent development streams of Laboratory, Field, Stakeholder Relations, and the Business and Compliance Case. We provide a graphic framework with criteria that must be met in order to proceed.

## 1. Background

*Aedes aegypti* and *Aedes albopictus* are invasive mosquito vector species that impose a substantial risk to human health. To control the abundance and spread of these arboviral pathogen vectors, sterile insect technique (SIT)-based approaches are a powerful complement to currently available methods. This technique is environmentally benign, specifically targeted, spatially constrained, and non-persistent, features which can help protect public health, non-target fauna, and the environment. Originally conceived for eradication of agricultural pest species, it was first proposed for vector species over 50 years ago [1] and using mosquito SIT as part of an integrated vector management (IVM) approach is now a reality for vector-borne disease control stakeholders.

The sterile insect technique disrupts the target organism’s reproductive cycle. Mass-reared males, sterilized using X-ray or gamma-ray ionisation, are released and may then mate with wild females, resulting in inseminations that do not produce progeny. Since the 1930s [2,3], SIT has been progressively applied in agriculture, contributing to the management of at least 20 species of insect pests [4,5]. The first effective applications of SIT to mosquitoes were in the 1960s and 1970s with pilot trials against *Culex quinquefasciatus* [6] and the malaria vectors *Anopheles quadrimaculatus* in Florida, USA [7], and *An. albimanus* in El Salvador, Central America [8]. Development and improvement of the technical steps have led to international interest in using SIT against some major vector species of *Plasmodium* spp. (malaria) (*Anopheles arabiensis*) and dengue virus (*Ae. albopictus* and *Ae. aegypti*). Pilot trials have now been performed on several continents [9], though few have yet been fully reported in peer-reviewed literature [10]. A recent field trial using a combination of SIT and the insect incompatibility technique (IIT, using the bacterium *Wolbachia*) succesfully reduced populations of *Ae. albopictus* in the residential areas of two small islands in Guangzhou, China [11]. In contrast to the long-established use of SIT in agriculture, application against human disease vectors requires more attention to the social perspective [12], mainly because pilot trials in urban areas need communication with multiple stakeholders and residents. Moreover, their direct interest in contributing to the vector control effort may appear less obvious to residents than to growers for whom yield and income depend on pest management success. Many lessons have been learnt [9], although further small-scale tests in a range of representative field situations are needed to demonstrate the wider operational efficiency of SIT for reducing mosquito populations below nuisance levels. Larger and longer-term pre-commercial programs will then assess cost effectiveness and impact on pathogen transmission thresholds. We describe here the crucial steps in planning and performing an SIT pilot test, while also setting the stage for future large-scale implementation.

Programs or stakeholders considering SIT as a contributor to their mosquito control activities will need local field pilot trials to demonstrate, evaluate, and calibrate effort in their specific context. Pilot trials can evaluate effectiveness and provide a first estimation of cost efficiency but also develop capacity for local SIT product delivery, for example, training of staff in new techniques and competencies, and engagement with public perception. Pilot trials can identify logistical, technological, and financial constraints to designing successful, longer-term wider-area IVM strategies. The explosive growth in scientific articles on specific steps in the SIT development process against mosquitoes is guided largely by specific research questions (reviewed by Lees et al. [13]), but a practical overview of pilot trial design and implementation is needed. Here we provide an accessible, harmonized, and structured ‘guide to process’ with subsets of conditions, based on key criteria, which must be met to support a journey towards well-conducted SIT trials (Figure 1). As part of the the European Union’s Zikaliance project [14] subtask on innovative vector control solutions, we hope specifically here to assist novice practitioners overcome potential problems, avoid wasted time, and increase their chances for successful control of *Aedes* spp. and *Aedes*-borne diseases.

The operational implementation of SIT is a phased conditional pathway (outlined in [15]): (i) determining feasibility, (ii) pilot testing, and (iii) operational/commercial program. This review describes the underpinning steps of the demonstration stages and the sequence of concurrent activities needed to determine feasibility (Figure 2), which largely focusses on developing technical capacities, funding model and stakeholder relationships, to the pilot trial. It does not cover operational phases or provide the advanced technical guidance covered by the International Atomic Energy Agency (IAEA) protocols [16] and the Joint IAEA and World Health Organisation (WHO) guidelines [17]. We provide, for a broad range of interested parties, an experienced, consensus viewpoint of ‘how to start’ with key go/no-go criteria. The features of SIT development critical to success are: stakeholder engagement at all levels; suitable pilot study sites; knowledge of target species dynamics and distribution under local field conditions; biological and technical requirements (including sex separation) for mass-rearing and sterile male releases, in addition to the design and evaluation criteria for small-scale field testing of sterile mosquito releases within IVM (Figure 1). This paper thus provides an accessible and compact document that is a pragmatic synthesis of the steps towards SIT pilot demonstration and is a support to early decision-making for the prevention of mosquito-borne diseases and nuisance control.

## 2. Opening Steps—Determining Feasibility

Several work streams will develop alongside one another as the project progresses (Figure 2), but it is important to first look carefully at the feasibility at the planned location.

### 2.1. Stakeholder Mapping, Regulatory Framework and Approval

An important early step is to ‘map the stakeholders’, that is, to understand who should be included in the project’s dialogue and development, and consider how and when they can be engaged (Figure 2). The first step in this, however, is to obtain outline approval or consensus at the highest level of the relevant authorities. 

Ministry-level stakeholders are likely to be aware of SIT, but policy-makers at local (regional or prefectoral) levels may appreciate background information. Communicating the historical context and national regulatory status (if any, or the regulatory approaches in other countries) along with outline information about the specific SIT pilot trial and the potential major application of its outcomes is important. The interest and acceptance of these decision-makers and establishing the potential for funding are pre-requisites for the development of a sustainable program. These conversations should use language appropriate to the recipient; it must be accessible, informative, and not overly-detailed and cautious (i.e., not overstate or propose a specific effect size for a trial). In circumstances where a potential trial area is identified early, having prior municipal-level support will help with approaching funders and policy-level actors. This is also a prerequisite for undertaking costly and complex actions, such as detailed entomological surveys or extensive stakeholder engagement at a site (Figure 1).

The regulatory context for SIT varies between countries and a clear framework is not common. There are no specific international regulations governing the use of radio-sterilized insects for pest control, although the International Standard for Phytosanitary Measures (ISPM) No. 3 [18] provides useful support and indicates requirements and standards for exporting, shipping, and releasing biological control agents and other beneficial organisms, such as sterile insects in agriculture. The recent common guidance by WHO-IAEA for SIT as a vector control solution [17] can further help public acceptance [19,20] and official support. From the outset it is important to determine which local and national authorities and regulations govern the release of sterile mosquitoes, because lack of clarity can lead to delay. In Brazil there were delays of two years [21] and in Reunion Island of five years before authorization to release was given [22]. In Europe, SIT programs using radio-sterilized male mosquitoes have been performed in the absence of a specific official regulatory framework but after validation by local authorities in Germany, Greece, Italy, Montenegro, and Spain [23]. 

Although SIT is generally accepted in the agricultural sector, human-health related interventions often require a consensus that matches social perceptions with scientific evidence. Establishing an advisory forum with locally relevant stakeholders is a useful relationship management strategy and provides a route to voice opinions. The stakeholder groups represented may include other pest control practitioners, expert scientists (entomologists, ecologists, epidemiologists, sociologists, mathematicians), religious groups, environmental associations, local politicians, local health professionals and services, and other interested parties such as school teachers or community groups. Encouraging two-way interactions with these groups ensures a legitimate approach. All key local players must have accurate information and understanding so that public dissemination through these trusted channels is correct and useful. 

### 2.2. Messaging Content and Progression

The engagement content follows a typical progression. Creating a profile for the problem and challenge faced (disease risk, nuisance, or both) helps when it comes to proposing unfamiliar technological solutions. The information should start with the biology and ecology of these urban mosquitoes and then describe the limitations of current traditional vector control methods. It is only later when uncertainties about the project are dispelled (plans, sites, and political support confirmed) that introducing the concept of SIT becomes useful [12,13]. 

At all stakeholder levels, transparent and participative decision-making processes are important for the success of integrated vector management (IVM) strategies. Local engagement is crucial for supporting IVM (e.g., by raising awareness of breeding sites and their reduction) and thus improves later likelihood of success of SIT. Useful information is available from the World Mosquito Program (formerly Eliminate Dengue) in Australia [24], where the engagement framework was designed to integrate the community with the program and increase their familiarity with the technical and research components. Early perception surveys among residents in La Reunion demonstrated that providing information several times worked to increase the social acceptance of SIT there (see [22]). 

#### Understanding Stakeholder Concerns

Suppressing vector populations through a biocontrol approach that reduces the use of chemicals will have positive impacts on health and the environment [25], although this can raise questions about potential unintended effects and teams should be prepared for this. The main concerns generally expressed are: (i) the risks associated with accidental releases of sterile females; (ii) the risks of released males not being fully sterile; (iii) the effects of releases of large numbers of individuals on associated fauna (predators and competitors for food resources); (iv) changes in species’ abundance and interactions via reduced larval competition; (v) the potential for population replacement by an alternative vector species in the vacant niche; and (vi) the potential for ionizing radiation contamination by released males. 

Where *Ae. aegypti* or *Ae. albopictus* are recent invasive species, it is unlikely that their suppression would negatively affect local biodiversity. Where these species are endemic or naturalized, the demonstrated wide use of SIT in agriculture helps provide reassurance [25]. The Organization for Economic Co-operation and Development (OECD) consensus document on the biology of *Ae. aegypti* [26] describes the ecological role of *Aedes* mosquitoes in the environment and also discusses biosafety issues arising from certain vector control practices. Environmental guidance documents, based on ecological evaluations of *Aedes* suppression such as those developed for *Anopheles gambiae* [27], are needed, although the biology of these species is well described [26]. While there will be local considerations, these would provide a comprehensive framework and inform planning of specific local ecological studies. Transparency throughout the process helps gain the stakeholder support needed for open-field trials [28], and the IAEA supports this with answers to many common questions [29].

### 2.3. Ethical Approvals

Pilot testing a new vector control strategy in urban and peri-urban areas often needs specific ethical approval for the pilot study [30]. This can be given by an institutional ethics board or other appropriate committee and, if applicable, by any relevant national or regional board. The community engagement plan can be used to demonstrate accountability to ethics boards by having appropriate and relevant communication, and associated risk assessment, based on an inventory of identified plausible risks [28,31]. There are particular ethical considerations for trials taking place in regions where education and literacy may be limited and communication plans are vital [32]. 

### 2.4. Financial Planning

To justify a pilot trial, decision makers and potential funders are likely to need an estimate of the long-term costs and benefits expected from a successful operational SIT implementation. There are numerous pest cost–benefit examples to support this [33]. Initial predictions can use econometric studies which report past and current costs of vector control measures [34,35], the spend on protection against mosquito bites by households [36], the impact on the tourism industry [37], and labour or social costs of disease cases for the country or region, such as disability-adjusted life years (DALYs) [38]. For example, in La Réunion, household-level expenditure on protective measures in 2012 was estimated at circa USD 28 million [36] and in Malaysia in 2009, the cost of dengue was estimated at USD 102 million (direct medical costs and costs related to lost productivity and premature mortality) [39]. Within the past decade, the global economic costs of invasive *Aedes* and their associated diseases are variously estimated to be at least USD 9–60 billion per year, though these are considered to be seriously underestimated [40,41,42]. Pilot trials will always be less cost-effective than large-scale implementation, where economies of scale will help reduce the unit production costs and female immigration from non-treated areas is reduced.

Estimating the costs of a pilot trial is difficult before the workplan and resources required are drawn up, and will depend partially on locally available skills and infrastructure, and already existing entomology surveillance. Estimating expenditure, even loosely, does allow useful comparisons with other control strategies and a wide variety of elements should be considered (Table 1 shows indicative components). A spreadsheet for the design of a mass-rearing facility to produce sterile male *Aedes* is available [43] to estimate the insect production component and can be used to compare with the costs of purchase from an established source. Mathematical models developed to estimate the costs of novel mosquito control technologies [44] can also be applied to SIT. The experience gained in costing and administering a trial will help improve long-term planning processes. Author experience suggests that in Europe where there is available expertise and practical support, preparation and trial can be achieved within four to five years for as little as EUR 200,000.

### 2.5. Aligning with Current Vector Management

The sterile insect technique is rarely a stand-alone tool and designing integrated vector management (IVM) approaches which combine sterile male releases with commonly used mosquito control interventions of proven efficacy is essential [45], both for pilot testing and for large-scale implementation (Figure 1). This creates the potential to reduce the target vector population in ways difficult to achieve with any single method and provides the basis for cost-effective and optimal use of available resources. Understanding and integrating with, or extending, the currently used tools in the area of interest is important as a springboard.

#### Combining Tools and Techniques for Integrated Vector Management

Traditional vector control methods (e.g., insecticide sprays and larval habitat reduction) are likely to be in use and locally familiar. In a pilot SIT trial they can be enhanced to reduce the local mosquito population, thus creating improved conditions for SIT, which is more effective at lower population densities. At the pilot scale, the release of sterile male mosquitoes is not a substitute for these common vector control activities. Maintaining these is important in the comparison between reference and treatment sites, and they are also the only ethically appropriate option if other vector interventions are already in place [46]. 

The individual effects of existing tools on vector density reduction are poorly understood and hard to measure [47]; reviewed in [48]. In addition, none of the ongoing and already reported small-scale field trials (e.g., Italy [49], China [11]) have an experimentally subset use of IVM components. Consequently, the specific combination of existing tools within any IVM package should initially be discussed with the current local vector control services.

Coordinating with local vector control agencies and, where appropriate, research institutions, in addition to communities, is important to make IVM more efficient, sustainable, and cost effective [48]. With some understanding of the local efficacy of specific methods, modelling can support planning of an IVM strategy and contribute to overall efficiency by optimizing the combination of control methods. Table 2 outlines some methods that may complement the deployment of sterile male releases, emphasizing an IVM approach with low insecticide use, but focusing on source reduction and mass trapping.

## 3. Selecting Pilot Trial Field Sites 

Well-designed studies demonstrating the local public health value of any unfamiliar vector control approaches are critical [46]. Iyaloo et al. [70] review the criteria for SIT ideal pilot site selection (Table 3), although finding sites that fulfil all criteria satisfactorily and that show initial similar densities of mosquito in intervention and non-treated areas is rare and there is likely to be some compromise.

Some initial and/or historical vector knowledge at candidate sites is essential. The site selection process should be collaborative and include the local and regional mosquito control services, SIT experts with experience in other locations, those who are planning production, vector biologists, and local authorities. This will help identify sites that offer a high probability of a convincing demonstration of mosquito population reduction. 

The pilot study should be considered as a possible starting point for an operational project, therefore, when possible, the pilot sites should be located in an area from which the release progression can be extended as in a rolling-carpet principle [4,15].

### 3.1. Site Size

The size for the pilot target area must provide an opportunity to successfully quantify the impacts of SIT [71]. Although there is no formula for calculating an adequate study size, extremely small sites (<3 ha) will not be convincing and excessively large sites (>20 ha) can be unnecessarily costly or time-consuming and can reduce the likelihood of success due to production and monitoring challenges. In general, the minimum area of the study sites should have a radius similar to the average dispersal distance of wild adults in that environment, but this may not be initially known. Experience suggests, however, that for an adequate demonstration of effect, a pilot field-site may be as small as 3–10 hectares.

### 3.2. Replication—How Many Sites?

The number and size of pilot areas will pragmatically depend on practical constraints such as suitable site availability, staff, and financial capacity (for detailed guidance in study design see [72]). Resources for surveillance, community engagement, and production are always limited and may be stretched in preparing for just two of the smallest sites available; this is a common starting point for nascent programs and most initial trials have no replication. The reference site(s) mosquito control activities will be managed according to the usual local vector control strategy and the test site(s) will receive sterile male releases as part of a more comprehensive IVM strategy [26]. 

The primary aims of the trial are to demonstrate that SIT can work in this location and that local operational capacity can be developed. Even with a single pair of reference-test sites, a random selection between these sites for sterile release allocation is advised. Where more than a single pair is available, contemporary data from two, or more, reference areas helps obtain a more robust comparison. Where there is sufficient time, reference and treatment sites can be alternated after two, or more, years. Because of inherent variability in mosquito population size due to temporal (seasonal) fluctuations and spatial heterogeneity due to landscape and host (human) characteristics, single-site studies without a contemporaneous comparison reference area (i.e., pre-post designs) provide lower-quality evidence. There may be longitudinal or spatial trends or other population-level changes not attributable to the intervention which can produce errors of interpretation.

More randomized, replicate sites reduce the risks of confounding and bias, and give improved estimates of differences in outcome between treated and non-treated areas (e.g., type of urbanization, socio-economic variables, human behaviour). This may be demanding for pilot trials as it requires more suitable sites that can be sufficiently isolated (minimizing mosquito migration), and sufficient resources to conduct monitoring activites and sterile insect production. Replicated studies are, however, the only means to demonstrate effect sizes in a statiscally convincing way.

### 3.3. Site Isolation

Among the most important criteria for an SIT test site is the level of isolation, be it ecological or artificially achieved, and a first screening for suitable sites can use satellite maps. The movement of adult mosquitoes from surrounding areas can confound data interpretation and prevent a solid, evidence-based trial outcome. This contamination can be caused both by immigration of fertile wild female mosquitoes to the test site from surrounding areas, and migration of sterile males out of the release area and into the reference site(s). 

When trials take place in a village or semi-rural setting, an uninhabited or ecologically inhospitable surrounding landscape may provide a natural barrier to migration for *Aedes* spp. If test and reference areas are within the same urban setting, however, they may need artificial isolation. There are a number of methods for reducing between-site contamination. One option is to ensure that clusters are separated by a sufficient buffer zone of approximately 2× mosquito dispersal distance estimates; several studies usefully estimate the flight distance of *Ae. aegypti* and *Ae. albopictus* females [73,74,75,76,77,78], although this varies according to landscape, natural resources, larval site distribution, climate, etc. [79]. Another option is to use a ‘fried egg’ design [4,15]; in this, continued releases of sterile males into buffer zones will protect the test area from invasion by fertile females. The buffer zones, although treated, are not included in the population monitoring and only serve as protection for the test area. This can be demanding for mosquito production. A further option is to install dense barriers of lethal traps or ovitraps, but the width and actual efficacy of such barriers must be determined in context [80]. In extreme cases, barrier zones have been produced by a combination of predatory fish, intensive source reduction, and adulticiding [81].

### 3.4. Presence of Vector-Borne Disease

Working in a area where endemic mosquito-borne diseases are common may pose ethical and communication challenges. Disease cases in the pilot area, which may have been imported through human movement, could affect perception of the success of the strategy and, in an epidemic situation, substantial adulticide treatment could be required. Passive transportation of mosquitoes (e.g., by cars) can be important dispersal routes for *Aedes* species [82] and could contribute to maintained pathogen transmission. It may be possible to select an area free of disease transmission, although, pilot sites should also represent the regional environment to which SIT would be operationally deployed. Where there is local transmission, stakeholder messaging must be careful not to create an expectation of epidemiological outcome from the trial and emphasise that current vector control should continue during the trial period. 

## 4. Characterizing Study Sites

Once the candidate sites have been identified, there is a period of characterization before a trial can begin (Figure 1). This consists of two parts, the human component, and the entomological component, and unless the area is unusually well-studied and baseline entomological data exist already, this will take at least a year.

### 4.1. The Human Component: Community Engagement

Locally relevant authorities should be contacted for their approval and careful engagement with the community should begin to present the plans, hear and understand their concerns, and ensure that expectations for the project are clearly communicated [83] (Figure 2). This will include a detailed local plan for integrated *Aedes* management (including larval habitat reduction), surveillance (ongoing mosquito abundance monitoring), anticipation of between-sector collaborations and supporting activities together with social engagement and ancillary aspects of vector control [48]. The stakeholder/community engagement plan will initially be similar between sites, though once these are confirmed as suitable and the randomization of treatment has happened, the communication will need to be shaped slightly differently in the test and reference sites.

When first communicating about a specific trial, a perception study of various segments of the populace will help to understand how the information is being received and will record possible questions, concerns, expectations, and misconceptions. This characterization of local people can help shape key messages addressed to those with reservations. Translating the technical concepts in a way that can be understood by lay people, however, is not trivial and is culture-dependent [12,48]. Willingness to be involved may vary according to the cultural and social context; this type of survey helps to plan the level, frequency, type of information, and method of information transfer, required. For any vector control intervention, a community or public acceptance plan aims to achieve an informed and accepting population at the field sites [30]. 

#### Information and Acceptance at Local Scale

The term ‘informed’ means that the people living at the field sites, and often the neighboring population, have been provided with information about the trial in a locally appropriate and accessible way. This can entail the development of locally-specific glossaries and engagement practices that ensure concepts are understood and sufficient inclusion is achieved [84,85]. For SIT against mosquitoes, the information should include: the basics of target mosquito species biology and ecology; the role of this species in nuisance and in disease transmission; challenges of mosquito control in urban areas; what is SIT; what is the plan of activities for the pilot study; what are the short- and long-term expectations of the trial; who is funding the project and who are the collaborators [29]. The Life CONOPS website [86], which provides information about a recent field trial in Greece, gives excellent examples of this.

Acceptance is a key concept in a pilot study and an informed population can give acceptance via the agreement of their elected or designated representatives. Consent is different and can only be given by individuals for a specific item or topic [87]. In this context, consent may be required for placing monitoring equipment in, or adjacent to, people’s homes or for gaining access to property in order to reduce mosquito larval development sites (source reduction) on private property (door-to-door campaigns). 

### 4.2. The Mosquito Component: Entomological Characterization

The local mosquito population must also be well-understood to guide trial design and this understanding is based on occurrence, abundance, and spatial and temporal distribution data. Preliminary sampling at periods of known high density can help with initial site choice, but an annual cycle, ideally more, of baseline entomological data for potential sites will confirm final selection (Figure 1). These data will provide detailed information on: (i) the level of isolation, (ii) the density of adult mosquitoes, (iii) seasonal variation, and (iv) the similarity between the sites that could serve as test and reference zones. With sufficient data from a test and reference site, the comparison can be made both within (pre-release year vs. post-release year) and between (test vs. reference) sites. 

#### 4.2.1. Baseline Entomological Data Collection

For SIT, the entomological data must be able to evaluate the fluctuations of adult mosquito densities, and once releases have started, the ratios of sterile to wild males, and natural variation in fecundity and fertility. Here, we focus on planning and testing SIT against a single species, whereas a challenge for many tropical countries are co-occurring disease-transmitting mosquitoes. Sympatric occurrence of more than one *Aedes* mosquito species (e.g., *Ae. albopictus* and *Ae. aegypti*) adds complexity to trial design and raises questions about ecological effects on non-target mosquito species; not all situations with several vector species are suitable for SIT. In these situations, any pilot trial will need to anticipate changes in the ecology, species composition, and relative densities in response to change in density of the target species (Figure 1). No current mosquito SIT program aims to control multiple sympatric vector species, although this is now being done for some agricultural pests [88].

Sites must be characterized using several trapping methods to minimize any bias associated with a specific method. Vector control teams usually conduct *Aedes* surveys in urban neighbourhoods using standard methods to evaluate the conventional indices: House, Breteau, and Container Indices. These surveys provide information on the presence/absence of aquatic stages (larvae and pupae), their spatial distribution within the area, and the typology of potential breeding sites, and makes it possible to estimate population densities indirectly.

Trap location affects attraction and therefore the data collected. These must be selected similarly across sites to avoid introducing sampling bias. To optimize efforts and to ensure that population estimates are reliable, trap location should be refined during baseline monitoring. Statistical analysis of subsamples can determine the catch-reduction effect of increasing trap density or help to eliminate trap sites that consistently capture no mosquitoes [62,89].

#### 4.2.2. Trapping Devices for Population Monitoring

In general, sampling methods should be practical and standard. When only one species of *Aedes* is present, ovitraps are the simplest and cheapest method, and can be complemented by fortnightly or monthly adult collections. When more than one species of *Aedes* coexists in the same area, then species identification of the collected eggs or resulting larvae is essential. In this situation, the entomological data collection should focus on regular (weekly) adult trapping, which allows rapid species identification. Less-frequent (two-weekly or monthly) ovitrapping can then be used, and a subsample sent for fertility assessment and species identification. Fertility data are used to assess SIT efficiency during the release period and baseline field fertility data allow technical practice and protocol validation. 

For *Ae. albopictus*, ovitraps should be placed in gardens, in shade or low vegetation, and preferably sheltered from wind and sun, most frequently around homes. *Aedes aegypti* trapping is also usually done inside houses; the ease of access and disturbance to residents must be considered and sites close to houses can be preferable. To lower the risk of interference, these are best-placed avoiding premises to which adulticide treatments are applied.

Adult mosquito densities can be assessed using various traps; BG Sentinel® odor-baited traps [90] are widely used for sampling *Aedes* globally. However, these require a power source and may not be suitable for all contexts; other trapping systems such as Adultrap [91] or sticky traps [92] can be alternatives. Except for male-only mark-release-recapture experiments [77], the baseline entomological data collection assumes standard, or locally determined, sex ratios; therefore a sex bias in trap attraction is a lesser issue. 

Trap density will depend on the study site area, but a minimum of 30 active traps (i.e., attracting mosquitoes during the preliminary survey) per site is needed to collect sufficient data. For large sites (>20 ha), cluster monitoring can be used; clusters of traps within randomly chosen blocks allow estimates per block and comparison between blocks within a study area (i.e., periphery vs. center). 

#### 4.2.3. Density and Dispersal Estimation: Mark Release Recapture Studies

Baseline sampling should be combined with mark–release–recapture (MRR) studies performed three or four times a year at expected highs and lows of mosquito abundance. These studies estimate adult population size by releasing marked mosquitoes followed by collections that include both those released and the wild population [77,93,94]. The ratio of released adults to total adults collected in traps allows estimates of absolute population size. Dispersal distance can be estimated from the distances travelled by marked mosquitoes from the release point. With these studies, the ethics and safety of releasing non-sterile adults in an urban environment must be considered; releasing females is widely thought to be undesirable [95].

Contemporaneous comparisons of the MRR population estimates and population indices from ovitrapping and adult trapping will allow future use of index data to estimate absolute population sizes [93,96]. Additionally, MRR can be used to evaluate site isolation and at later stages the mortality and competitiveness of the sterile males [97]. 

The reliability of MRR-derived estimates depends on the intensity of collection efforts. These vital studies must be effectively planned to have sufficient numbers of male mosquitoes to release (>5000 males per release station), and to have sufficient traps and human resources to perform the collections. Practice with MRR helps to train staff, assess requirements, and refine the local MRR design, and will be helped by the established monitoring network of adult traps. Alternative adult collections for MRR can use human landing catches [98], although these target female mosquitoes and are not recommended in areas where disease transmission is known to occur. Other methods such as back-pack aspiration of eaves and other resting habitats or ‘mosquibat’ sweeping sticky rackets may also be used [96]. The advantage of these methods is their mobility, which can allow coverage of larger areas and a focus on sites where mosquitoes are usually found (near tree trunks, in shaded areas, under vegetation, near fruit for feeding); they require standardized protocols to reduce sampling biases from operators. 

## 5. Mosquito Sourcing: Purchase or Production?

The number of sterile males required for release can be substantial and two options exist: local production and purchase/importation. If local infrastructure (insectary and irradiation source) is unavailable or too expensive, acquiring sterilized mosquitoes from an established facility, sometimes abroad, is possible. Each route has both advantages and disadvantages (Table 4) and the United Nations’ Food and Agriculture Organisation (FAO) or the IAEA can serve to introduce interested parties to established production facilities. It may be recommended or required to use a local mosquito strain for a pilot trial, though this might not be applicable to an operational area-wide phase, or to countries relying on another country’s production capacity. 

### 5.1. Local Sterile Male Production

One of the major challenges at SIT pilot scale is the ability to establish a reliable and efficient mass-rearing system that can produce a sufficient number of sterile males to impact the target population. Mass rearing is not a trivial undertaking and any SIT program using local production must solve ‘input questions’ such as how to rear larvae and adults at high densities or how to optimise egg production from adults. Output measures to consider are not only the quantity but also the quality of the mass-produced sterile males [99]. Routine quality estimation using proxy measures, such as pupal size, adult survival, flight ability and mating capacity, must take place regularly.

#### 5.1.1. Rearing Facility

Rearing facilities must continuously provide sterile males for the duration of the release phase. It is also important to consider the scale up of mosquito production to the level of anticipated future release numbers (from pilot study to field implementation) at the design stage, including the possibility of a modular, expandable facility. There is an initial learning curve and mosquito production will increase with rearing experience, mechanization and sexing efficiency. The construction and operation of mass-rearing facilities for SIT may represent a major investment and requires strong technical, political, and financial support for success. Detailed guidance for a mass-rearing facility with a worksheet for scoring pertinent factors is available [43]. 

#### 5.1.2. Mass Rearing Process

Significant advances have been made in mass-rearing technology, formulation of mosquito diets, and sterilization using precise and uniform irradiation dosage. Prototypes of mass-rearing equipment (racks, breeding trays, mass rearing cages) have been developed and tested. The standard operating procedures (SOPs) for routine rearing and quality control produced by the FAO/IAEA can be easily adapted [16] and the mass-rearing systems available are easy to use and can produce up to 900,000 pupae per rack/week for *Aedes* species [100,101]. 

#### 5.1.3. Sex-Separation Systems

One criticial step for mosquito SIT is an efficient sex-separation method, because the release of females has to be avoided. Currently for *Aedes* spp., which are sexually dimorphic, this mostly relies on mechanical separation techniques, which can lead to 99% male purity [102] but are highly operator-dependent. The development of a genetic sexing system (GSS), e.g., the separation or killing of females enabling the production of a male-only population [103,104], or other innovative methods to improve sexing accuracy would be a valuable asset to the *Aedes* spp. SIT package [105]. An automatic sex-sorter developed by Wolbaki is being tested at the Insect Pest Control Laboratory (IPCL) of the IAEA, and estimates a female contamination of <0.3% at a rate of up to 150,000 pupae/hour [106].

#### 5.1.4. Sterilization 

Sterilization is performed during the last developmental phase, at the earliest on pupae aged >24 h, when the organism is less affected by the radiation (i.e., when most of its somatic cells have completed their development and only the reproductive cells are altered) [107]. However, variation in growth rate and in the optimal window for pupae collection inherent to a larger-scale rearing process can lead to a significant loss of males. The sterilization of males as adults is more efficient in terms of yield and labour for small-medium sized programs.

The doses necessary for complete or partial sterilization of male mosquitoes must be established for the release strain using the irradiation source available for the pilot study. The dose required depends not only on the *Aedes* species but also on several technical parameters including the radiation source, dose-rate, and container [108]. The effectiveness of SIT depends on both the competitiveness and sterility of the released males. High radiation dose and handling practices are associated with reduced vitality of males resulting in lowered mating competitiveness; the right dose is that which maintains both competitiveness and efficiency [109,110,111].

### 5.2. Quality Control

Whichever path is taken to source the sterile males, quality control is essential. Many aspects of production can affect the quality of the males: diet quality and quantity, artificial environment conditions, genetic drift, rearing densities, handling during separation, transport, and sterilization. Important data on the mating competitiveness, flight ability, survival, and dispersal of the sterile males used should be gathered during the preparatory phases (Figure 1 and Figure 2). Detailed information on these various procedures is found in the IAEA/WHO guidance document [17].

Some technical standards, such as the level of residual fertility allowed and the resulting potential risk of introducing undesirable alleles (such as insecticide resistance) to local populations, or the percentage of female contamination allowed in the male releases and their vector competence risks, may be pertinent. Estimation of these should be discussed with the relevant authorities and scientific experts. 

## 6. Trial Implementation

### 6.1. Enhanced Stakeholder Engagement

Initiating the trial phase requires maintaining active support and participation of multiple stakeholders because, even if the trial is local, awareness will be area-wide (Figure 2). This also anticipates future application in wider and different habitats and for different purposes, including disease prevention and community well-being. Enhanced stakeholder engagement at the trial sites increases effectiveness and information should be delivered through many means. Prior engagement has set the scene and it is now the time for reinforcing an accurate description of the release and monitoring activities and their objectives. This can be done through town hall meetings, delivery of leaflets, and face-to-face methods, depending on the size of the sites.

Although sterile male releases will be performed by project staff, source reduction, larvicide application, and the use of adult traps may involve local vector control services and/or private control companies. The participation of local residents helps to reduce background mosquito populations, but can be hard to motivate especially if mosquito numbers fall. Participation of the vector control services in source reduction on private property can be an incentive to stimulate public participation, a route for engagement or act as enforcement through regular inspection [53]. In some situations, however, it may be viewed as a sufficient intervention by local residents, reducing their own responsibility [112,113]. Maintaining an open and communicative relationship with all stakeholders helps to foster participation and supports the delivery and sustainability of the pilot deployment. 

### 6.2. Optimizing the Release Strategy to Meet Objectives

In any SIT pilot trial it is essential to define the aim of the interventions (reviewed in [4]) and to estimate the magnitude of the effect and predict the likelihood of success. These enable calibration of sterile male production and design of the release strategy, and allow for effective evaluation. Although SIT can be also applied to prevent the invasion and establishment of an invasive population into a new area or to contain its introduction, the suppression goal for an established population is most often to maintain this below a specified threshold. This then reduces the biting nuisance to an acceptable level, and may ultimately suppress disease transmission. In certain contexts, local eradication of a population could be the aim; this may apply more easily to recently invaded areas or island settings where initial mosquito densities and immigration are low.

Predictive mathematical models help define an achievable suppression level and design the size and types of releases (continuous, periodic, or pulse) [114,115]. They will be based on parameter estimates from baseline entomological field data (density, site size, and isolation level), the performance characteristics of the sterile males (level of sterility, competitiveness, survival, and dispersal capacity), and the production capacity. 

#### 6.2.1. When to Start?

Releases of sterile males have the strongest suppression effect when the numbers of released males overwhelm the wild male population, so releases should start when the target species density reaches a seasonal minimum or when mosquitoes have been temporarily reduced by a pre-release suppression phase [116]. In some regions, *Aedes* mosquitoes show a marked seasonal fluctuation in population density [117], more clearly so in temperate areas [89,118,119,120]. 

#### 6.2.2. Release Ratios

The sterile-to-wild male release ratio in sterile insect programs against agricultural pests can vary from 7:1 (for some tse-tse fly sub-species) to 100:1 (in some contexts for screwworm) [121]. The appropriate release ratio depends on several factors, including the spatial distribution of the wild population, the sterile male dispersal, survival, and mating competitivity. For mosquito SIT trials, a release ratio of 10:1 is generally the minimum necessary to maintain a high likelihood of wild females mating with a sterile male. This ‘guidance’ ratio can be adjusted by ongoing monitoring of local densities and as the release intervention progresses. With constant release effort, this ratio will increase over time if the SIT is efficient in reducing the wild population.

#### 6.2.3. Release Frequency

The optimal release frequency aims to maintain at least a minimum of the desired sterile to wild male ratio during the release period. Understanding the daily mortality of released males is thus important and will come from the early MRR studies. In agriculture, releases are generally performed weekly or biweekly. A recent mosquito SIT (combined with IIT) in China [11] used a release ratio of 5:1 three times per week. Practicalities will also influence this decision and it may not always be cost-effective to release sterile males continuously/daily; instead teams can use less frequent but more intense pulses [114].

#### 6.2.4. Release Locations

The spatial distribution of ground-level release stations is informed by the mean dispersal distance of male mosquitoes estimated by the earlier MRR studies and by the patterns of relative density established through baseline monitoring. Male *Ae. albopictus* typically travel 100–150 m [77,93,122], but have been found 1–2 km from their release point [122]. There are sex differences in dispersal in some environments, with most females found within 250 m of their release [122]. Both sexes of *Ae. aegypti* typically disperse less than this and most estimates suggest 150–250 m is typical [123,124,125,126]. 

Early trials released male pupae [49], although more recently young adults have been used [11] because remaining in the insectary with access to food for a few days improves competitiveness and survival [127,128]. More dispersed, aerial releases of adults using drones can provide effective and even coverage of larger areas [129]. 

#### 6.2.5. Marking Mosquitoes

Most agricultural pest control programs release color-marked insects to distinguish sterile males during field monitoring and in MRRs [130]. This is not ideal for mosquitoes because the fluorescent color might create anxiety in people. The dust may also alter behavior and is lost with time, making it difficult to estimate the competiveness of the released males. The use of internal marking also detectable in the semen (e.g., rhodamine [131,132], or a stable isotope [133]), may become alternatives and allow both identification of released males and their contribution to the insemination of wild females. Other innovative methods, including liquid-applied fluorescense and synthetic DNA-tags, hold great promise for future studies [134].

### 6.3. Pilot Trial Duration

*Aedes albopictus* populations present a particular challenge as the immediate effect of SIT can be buffered by their ability to diapause. They can produce quiescent embryos that may hatch weeks or months after oviposition. These embryos are insensitive to SIT and releases may need to be performed for at least two seasons to be able to clearly identify effects. 

Achieving the necessary production and refining release logistics are likely to take several months and the full potential of pilot trials may not be realized in the first season. In some situations, such as where the site is well isolated and has initial low-density levels of mosquito population, a single season may be able to achieve identifiable reductions. Suppression obtained during the first year will led to increased effectiveness subsequently, therefore a pilot program of 2–3 years or seasons is desirable.

## 7. Pilot Trial Evaluation

### 7.1. Defining Success

In most cases, the ultimate objective of using SIT against *Aedes* is to maintain low mosquito population density in a sustainable way and, where relevant, to reduce or prevent arbovirus transmission. Where disease transmission is not an issue, the reduction target may be to a density that is socially acceptable (e.g., in touristic areas). During a pilot trial, however, the objective is to demonstrate if, and how, a reduction of the target population is possible. It will be difficult in an SIT pilot trial against *Aedes* to demonstrate an impact on *Aedes*-borne disease transmission. People move around widely and the likelihood of being bitten by infected mosquitoes outside the test area remains the same, irrespective of vector suppression at the test site. This is a question of scale. Because pilot tests are carried out on a relatively small scale, only entomological indicators are appropriate to evaluate the outcomes.

Later stages may move to larger scale trials, and these can evaluate the effect of SIT intervention on reducing the diseases transmitted by *Aedes* spp. mosquitoes (e.g., a seroprevalence survey in communities with and without SIT) using guidance published by WHO and IAEA [17]. Island settings where human populations move little and where the entire island could be covered by release of sterile mosquitoes might provide a situation and an opportunity to reliably estimate the impact of SIT on epidemiological outcomes.

### 7.2. Evaluation of Field Release Efficacy

Suppression in population size is usually measured through a sufficient reduction in population size relative to pre-release and/or comparison site densities. Comparability is improved if the natural temporal and spatial variation is controlled for through replication, longer studies, or site-switching (see Section 3.2). Statistical models informed by baseline site characterization data can help to predict the reduction in population size likely to be detectable using the data that are routinely collected (via power analysis). For all measures, geographic information systems (GIS) and accessible spatial data are increasingly being used in vector management programmes to support field population evaluation [135].

The impact of sterile male releases can also be assessed by measuring the proportion of egg sterility in eggs from ovitraps. A net induced egg sterility of 80%, achievable with sufficiently high release ratios, may be a good target to deliver a clearly observable reduction in the adult population. Further comparison of wild male/female captures between the release and control areas will give date-specific estimates of population reduction expressed as 100% × (1 − wild mosquitoes per trap per day in treated site/wild mosquitoes per trap per day in non-treated site). 

When mosquito densities are low, conventional entomological indicators may not be suitable for evaluating efficacy due to large variation in trap catches. In these cases, nuisance and biting rates may be useful indicators. Salivary antibody-based biomarkers can also be used as secondary indicators of a reduction in biting [136].

### 7.3. Capacity Development

Another objective, sometimes lost from view, is that of capacity development. Staff in all trial components from stakeholder engagement and insectary management through to field monitoring all develop their skills and contribute this capacity to future vector control initiatives. This has great value, particularly when the initial experience of the teams is limited. The success of many SIT programs depends on overcoming unanticipated challenges, such as lower/higher than expected mosquito populations, high levels of immigration or mating failure, and inefficient initial reduction of the wild population. The ability to produce and release a high quality sterile mosquito does not mean that the programme will be successful. The quality of the technical aspects and the quality of the implementation strategy are equally important, and both will improve with staff experience, and time (Figure 2).

### 7.4. Evaluation of Production and Release Quality Control

The competitiveness of a sterile male is the likelihood of a wild female mating with a sterile male when sterile and wild males are present in equal numbers, and continuing efforts must be made to improve the competitive quality of sterilized males produced in any mass-rearing system. During a trial, indirect measures of sterile male competitiveness can be made through MRR experiments using marked sterile males, and relating the sterile-to-wild male ratio to the observed egg sterility level in the field. This type of evaluation should be conducted regularly and especially after any change in mass rearing procedures, transport, or release procedures. Results will help to identify issues and permit adjustment of the release strategy in the field.

### 7.5. Feedback to Stakeholders

An important element of stakeholder relations is providing feedback on the progress and results of the trial (Figure 1). Local feedback at trial sites can be delivered informally and form part of regular data collection. If nuisance and biting rates have been recorded, then these provide immediate local feedback and could also be established as a metric. Health authorities and strategic stakeholders, need accurate, precise, and clearly presented analysis and evaluation. 

Proceeding with SIT depends in part on having the political and financial support needed to sustain the program until demonstrable reductions are achieved. One aim is often to produce preliminary data that informs future production and release capacity, and it is essential that this measure of success is also understood by relevant stakeholders. With appropriate expectations, local experience will grow and it is highly likely that the releases will be ‘successful’. 

## 8. Conclusions and Perspectives 

This review presented a practical and accessible introduction to using SIT against *Aedes* mosquitoes and, although an *Aedes* SIT package is now close to being ‘field-ready’, many steps and parameters within it will continue to need local adjustment and validation. For those who decide to progresss further down this path the Insect Pest Control Section|Joint FAO/IAEA Programme of Nuclear Techniques in Food and Agriculture has now published several detailed guidance documents addressing this process [137]. The integration of SIT into vector control practices should be guided by sound science based on a harmonized and structured process for data collection, interpretation, and reporting. In parallel to pilot testing, further important research must continue to improve future implementation.

The development of SIT for agricultural pests took decades before reaching a level of delivery and know-how that was efficient and cost-effective as part of area-wide integrated pest management approaches and the same will be true for SIT against *Aedes* mosquitoes. The aims of this review were to provide an overview and to support teams exploring the possibility of pilot field trials. Failure can be costly, and failure due to poor preparation and excessive haste might have long-term negative impacts. Here we hope to improve the chances of SIT success by providing a clear initial route map, and interested parties will build on this with collaborations and seek further guidance [138].

The outcome of a pilot trial can determine whether an approach integrating SIT will be granted local and regional public acceptance, and can raise the interest of decision-makers for future operational use. It is encouraging to see international organizations such as the WHO take a position acknowledging the role of SIT as a promising technology to be deployed against vector-borne diseases. Influential international institutions will also be pivotal in providing future guidance for critical components of the SIT research process and potential follow-up positioning on key issues, such as recommendations and policy advice, to support a more flexible regulation of SIT as part of the IVM approach for vector-borne disease prevention. 

## Figures and Tables

**Figure 1 insects-12-00191-f001:**
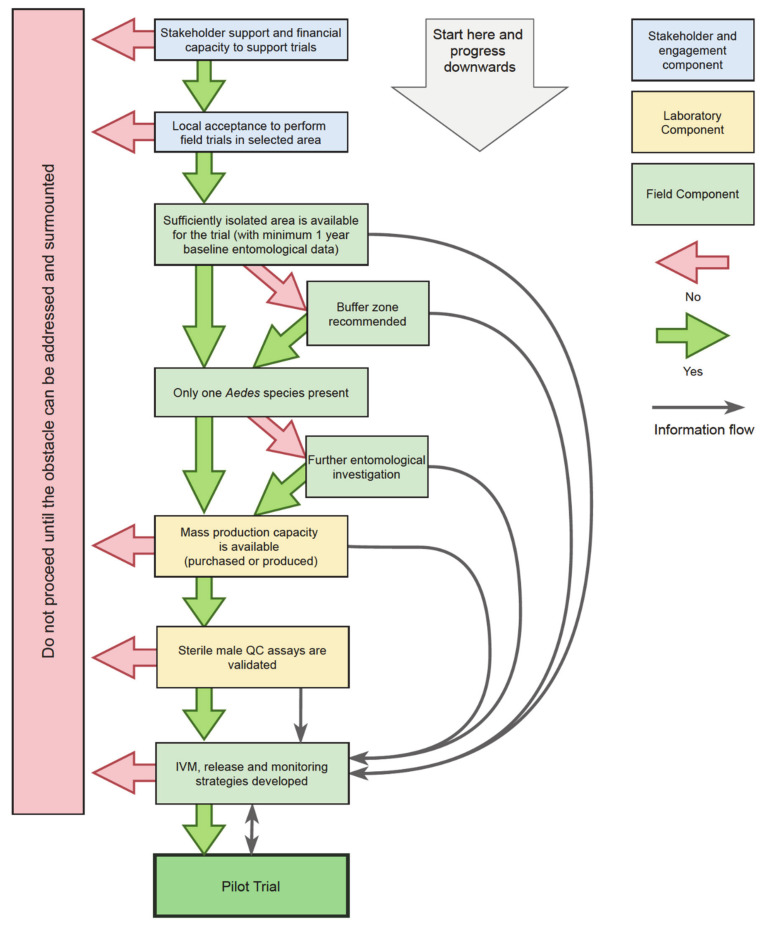
A decision tree for progression towards the implementation of a sterile insect technique (SIT) pilot field trial for vector mosquito management.

**Figure 2 insects-12-00191-f002:**
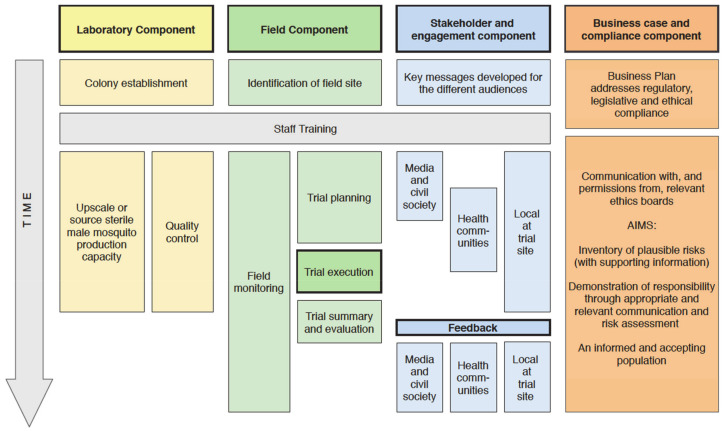
Illustrative timeline of the steps towards the implementation of a sterile insect technique (SIT) pilot field trial for vector mosquito management.

**Table 1 insects-12-00191-t001:** Pilot trial sample cost items.

**Stakeholder Engagement**	**Production**
Message development	Larval and adult diet
Media costs (electronic and printing) Social networking	Staff training and costs Sex sorting
Staff training + costs	Disposable materials
Transport and site costs	Sterilization (local or through purchase)
**Facility operation**	**Release and monitoring**
Facility leasing, renovation or purchase costs	Staff training and costs
Administration	Traps
Maintenance and utilities	Disposable materials
Cleaning services and staff costs	Transport and site costs
Waste disposal	Data management and analysis

**Table 2 insects-12-00191-t002:** Mosquito control methods compatible with sterile insect technique (SIT) of mosquitoes as part of an integrated vector management (IVM) approach.

Source reduction ranges from simply reducing artificial larval sites (removing or draining non-essential or disposable containers on most properties), to long-term habitat-altering measures, and may almost eliminate any need for larviciding. This approach, which primarily targets artificial containers in private and public spaces, is emphasized in any SIT pilot intervention. This requires the active involvement of the community and a public information campaign is essential to obtain community mobilization [50,51]. Larval habitat elimination through public engagement is useful, but rarely sufficient without the active involvment of supervision from official control services [52,53].
Door-to-door (DtoD) or house-to-house based reduction of larval habitats of *Ae. albopictus* has been used in several trials in Italy, and is both legally enforced and inspected as part of mosquito control in Singapore. This entails regular inspection of private properties, with larvicide treatment of permanent larval sites and the removal or treatment of temporary ones (source reduction), together with relevant control information provided to the residents. In this program, accessing 95% of the private properties, resulted in a 69–72% reduction in the density of *Ae. albopictus* females and a 36–62% reduction in ovitrap collections [54]. For effective and sustained source reduction, DtoD requires appropriate communication materials, planning and manpower.
Larviciding is the use of chemicals or biological agents to control immature stages developing in aquatic habitats. The feasibility, effectiveness, acceptability and cost of biological larviciding in some countries are summarized in Guzzetta et al. [55]. The most widely used tools are biological control agents such as copepods, *Bacillus thuringiensis israelensis* (Bti), *Lysinibacillus sphaericus* (Ls) and spinosad [56,57], or insect growth regulators (IGR) such as diflubenzuron, methoprene and pyriproxifen. Auto-dissemination (AD) of the pyriproxifen can deliver impressive levels of suppression in field trials [58,59]. However, AD works well at high, but not low, mosquito densities and some degree of population recovery is expected once pyriproxifen levels fall [60]. Area-wide application of pyriproxifen cannot provide long-term reduction in mosquitoes because cryptic populations are likely to re-surge; so coupling AD with SIT has great potential [60]. Larval habitat treatment with IGRs within release areas may also reduce the influence of any immigration by fertile females.
Physical control by gravid mass-trapping and kill tactics. Each gravid *Aedes* sp. female can lay 100–200 eggs and contribute to rapid population build-up with a generation every six to nine days for *Ae. albopictus* under optimal conditions. Originally developed for population monitoring, gravid traps baited with an oviposition medium and with either sticky devices or insecticides can reduce local population densities [61,62]. The use of three CDC autocidal gravid traps per home in more than 85% of houses within a treatment area has shown sustained and effective reductions (80%) of *Ae. aegypti* populations [63,64,65,66]. The BG-GAT (Biogents’ Gravid Aedes Trap) against *Ae. albopictus* has shown good results in the USA when house coverage is over 80% [67].
Ground or aerial adulticiding with ultra-low volume (ULV) spraying of pyrethroid or carbamate insecticides is usually employed during public health emergencies to reduce further human disease transmission. Depending on the *Aedes* species present, indoor or outdoor residual treatments can affect vector densities [68]. In some contexts, when the natural population of *Aedes* remains high all year long and where there is no evidence of insecticide resistance [69], adulticiding may be neccessary to reduce the wild population preceding the sterile male releases. This requires careful weighing of the risks and benefits in each situation. Insecticide resistance should be carefully monitored, not only in the pre-suppression phase, but in the broad context of any IVM strategy [61]. Long-term use of adulticides should be avoided because of stakeholder perception and the risks of resistance and potential impact on non-target organisms and the environment.
Indoor residual spraying using pyrethroid or other insecticide classes can reduce the presence of *Ae. aegypti* females because these prefer biting and resting indoors. This is showing promising results in tropical and subtropical areas of Latin America [69].

**Table 3 insects-12-00191-t003:** Important criteria for selecting pilot study sites.

**Entomological:** Initial or historical estimates of vector bio-ecology and density, validation of monitoring tools (traps adapted to the context). **Ecological:** The degree of ecological isolation.
**Logistic:** The expected sterile male mosquito production levels that would be required for the site, the constraints (geographical, topographical, etc.) related to mass releases.
**Social/ Financial:** The expected political/social stakeholder support available locally and regionally.

**Table 4 insects-12-00191-t004:** Advantages and disadvantages of local production and importation of sterile males for sustaining a sterile insect technique (SIT) pilot trial for vector mosquito management.

	Local Production	Purchase/Import
**Advantages**	Ease of logistics More control/freedom over rearing schedule Builds local capacity Better quality control	No need for substantial initial investment No need for irradiation source Efforts can be focused on field activities
**Disadvantages**	Initial investment for a sufficiently large insectary Irradiation source must be available and convenient Staff training costs	Possible loss of quality and competitiveness following long distance transportation Use of a local strain may not be possible (if the provider can not rear different strains) Importation permits required Logistic chain management Risk of gene flow to wild population if sub-sterile males from a non-local strain are released

## Data Availability

Data sharing not applicable.

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
