# Peer review of "Sterile Insect Technique (SIT) against Aedes Species Mosquitoes: A Roadmap and Good Practice Framework for Designing, Implementing and Evaluating Pilot Field Trials"

_insects, 2021, doi:10.3390/insects12030191_

Round 1
Reviewer 1 Report
Oliva et al. provide a roadmap for the design and implementation of pilot field trails to evaluate the sterile insect technique for control of Aedes vectors of arboviruses. This issue has gained increasing attention over the past decade or two and will be of interest to readers of Insects. The authors have considerable experience in this area.
The manuscript has no major flaws in my opinion and could be accepted with moderate revision.
Main points.
I have only two main points.
First, I think the ethical issues related to performing pilot scale studies in inhabited areas are mentioned in passing but are largely overlooked. The need to seek ethics committee approval is also mentioned briefly (section as if it were almost trivial. There are a number of issues related to the consequences of the accidental release of females, use of adulticides, potential introduction of insecticide resistance genes, informed consent by house owners for particular actions especially in regions with low levels of education and literacy, among others, that deserve closer examination, as many entomologists (i.e., readers of Insects) will likely be poorly informed or unaware of the ethical implications. The authors could point the reader to suitable sources that have addressed the ethical issues of mosquito releases in detail.
Second, as someone that has to write technical texts in another language I sympathize with the authors, but I did find the writing style hard going in parts. There are issues of sentence structure, word order and use of words that could benefit from careful editing by the native English speakers (MQB & CMC?). This would significantly improve the readability of the text, particularly in some sections.
Minor points.
I have written numbered points on a scanned copy of the manuscript (attached).
1. Is this an affiliation? More information required.
2. More information required.
3. I could not discover what TSUP Obras was in a Google search. In general, the affiliations are highly variable with many abbreviations and a variable format.
4. and 5. SIT was originally designed for pest ELIMINATION and has only been considered as a pest management tool more recently (last two decades?). Please make this clear.
6. Define SIT at first use.
7. I did not understand this phrase.
8. The authors appear to suggest that reduced chemical control will assuage stakeholder concerns on health and environmental impacts. However, elsewhere the authors state that use of adulticides to reduce target populations is desirable prior to male release and autodisemination of IGRs such as pyriproxyfen (a synthetic chemical insecticide, albeit of v. low vertebrate toxicity) can usefully contribute to vector control. The authors could clarify this apparent inconsistency.
9. Do stakeholders really express concerns about gene flow? I could see how concern may be expressed about the release of irradiated fertile males. Could you clarify or rephrase?
10. The issue of replication concerned me. The authors appear to suggest that it is usual to perform unreplicated trials in the early stages of SIT evaluation program, as long as an untreated control site is available. This is okay as long as the operational objectives are made clear in terms of overcoming logistical issues and to gain experience of production, QC, release and monitoring on a small scale prior to a replicated study. By their nature, unreplicated studies do not provide any scientifically valid information, although they may have anecdotal value. One way to mitigate such issues would be to switch (rotate) treatment and control sites over several seasons/years.
11. I think the authors should emphasize that replicated studies are the only means to generate sound science (highlighted on L713).
12. This observation applies to Ae. aegypti, but not Ae. albopictus, correct?
13. Please explain what this means.
14. The possibility of selecting truly "representative" sites that are free of disease transmission would appear to be very limited indeed. Better to emphasize that conventional vector control measures need to be undertaken in both SIT and control localities during the study period.
15. An issue of concern is that the mechanisms of information transfer are suited to the educational and literacy status of the local population - - particularly in developing regions, i.e. the regions in which most arbovirus transmission occurs.
16. Exactly, could you explain this and provide a couple of references - -there are ample examples of outreach programs with resource-poor farmers that could be adapted for public health purposes.
17. Please explain why several trapping methods are required.
18. Deleted in revision. Ignore this point.
19. What is a standard sex ratio?
20. Please explain/clarify what you mean by this.
21. Was this really the aim of the review?

Author Response
(purple is our response, red indicates an alteration to text or an included citation)

Reviewer 2 Report
The manuscript “Sterile Insect Technique (SIT) against Aedes species mosquitoes: A roadmap and good practice framework for designing, implementing and evaluating pilot field trials” by Oliva et al. aimed to summarize a comprehensive, step-by-step and multi-stakeholder approach for conducting relevant pilot SIT field trials. The manuscript is very interesting and timely and will serve as a guide for future SIT studies and implementation strategies.
The authors critically discuss the most important points for the implementation of a SIT program. However, the limitations of the SIT strategy as well as its effectiveness were not as thoroughly discussed as its advantages.
Some key limitations should be included in the manuscript to better inform scientists and stakeholders, including but not limited to the below:
Mass rearing millions of mosquitoes is not a trivial task and will require substantial investments and training. Furthermore, sex sorting males from females is not only laborious but half of the resources (needed to rear females that will be discarded) is lost.
Releasing irradiated mosquitoes also has limitations that need to be addressed. Fitness loss is a major problem is should be taken seriously in a successful operation. Moreover, different populations/species have different tolerance for radiation rendering results from other areas uninformative.
However, most importantly is the assessment of the effectiveness of a SIT release program by a randomized entomological trial to assess if the program has a significant impact on the mosquito population. If the results are promising a subsequent randomized epidemiological trial is needed since vector abundance and arbovirus incidence are asymmetric (reducing the vector population in half does not mean the incidence will also be reduced half). I would encourage the authors to include and discuss the VCAG recommendations.
Lastly, the cost is very important to any mosquito control operation. That has to be considered and further discussed.
Author Response

(The authors gave the same response as above.)

Reviewer 3 Report
The authors provide a lengthy technical manuscript that provides guidelines for the release of sterile male mosquitoes as a control methods for invasive Aedes.
The authors did a good job to cover all the practical steps that should be evaluated and implemented to lead to a successful mosquito control plan that integrates SIT.
While the technical aspects of the roadmap are well covered and explained, some important scientific facets are neglected and information is often anecdotal and not backed up with data or scientific studies.
Abstract: The abstract should be rewritten, the language is sometimes convoluted and sloppy (for example “at all levels in support of this”, this what?).
All throughout the paper: I would be sure to indicate that you are referring to “invasive Aedes” and not Aedes mosquitoes, which is a much more diverse taxon than “invasive Aedes” species.
Line 56 and throughout the manuscript: please be precise with the scientific language, “arboviral disease vector” is unclear. Generally, disease (conditions) cannot be transmitted by mosquitoes whereas pathogens (viruses) can.
Line 59: What do you mean with “ecologically benign”?
Line 59: “species-specific”
Line 63: This sentence is not clear, rephrase.
Line 70: What do you mean with “a graphic order of criteria”?
Line 80-81: This sentence needs citations as it strongly states some important facts which are taken for granted without providing supporting data.
Line 82-84: Same, add citations that support this sentence.
85: The principle? Isn’t the scope?
86: Are released and then they may mate with females, please state things correctly.
91: El Salvador is in Central America!
93: Vector of Plasmodium sp. and dengue virus.
106: a not an
203: This statement needs to be backed up with data or references in which the effects of SIT on health and environment were evaluated or tested.
214: This statement needs more information on what are the “demonstrated safety and efficacy of SIT”; moreover the sentence needs to be backed up with citations.
Box 2: Door-to-door: Please use data from “one of the several trials in Italy” instead of reporting anecdotal unpublished data.
Box 2: Reporting that traps and methodologies which were developed for surveillance and monitoring as effective control methods seems naive and misleading for the reader. There are a handful of studies which tested this possibility, however its efficacy seem to be extremely local and restricted in time. Consider to remove this section or to explicit the limitations of this approach.
289: What you mean with “is rare”? Please explain.
452: How did you derive the 30 traps per site? This figure seems to be thrown there out of the blue. Please back up with data, citations or at least explain how you decided that 30 is the best number.
596: Please back up with data or references the statement on the optimal ratio sterile:wild males.
613: These two statements are unsupported by recent publications on dispersal of both Ae. aegypti and Ae. albopictus. Please check the following publications and update the manuscript accordingly:
https://journals.plos.org/plosntds/article?id=10.1371/journal.pntd.0005347
https://esajournals.onlinelibrary.wiley.com/doi/full/10.1002/ecs2.2977
https://www.nature.com/articles/s41598-020-63670-9
623: Rhodamine B is bright purple colour and therefore may present the same problem (it is staining fluorescent dye) as the fluorescent powders. Apart from stable isotopes, check recent advancement in DNA marking for mosquitoes, for example:
https://www.biorxiv.org/content/10.1101/2020.08.23.262741v1
627: For my understanding diapausing is not a trait of the life cycle of Aedes aegypti as well as of some (tropical) populations of Ae. albopictus. Please be accurate in explaining to what species you are referring.
645: Will lead
653: sterile release? “release of sterile mosquitoes”
664: How did you derive 80% as a good target? Add data, citations or explain how you guessed this percentage.
Author Response

(The authors gave the same response as above.)

Round 2
Reviewer 3 Report
The authors did a great job in revising the manuscript, and spelled out some of the concepts and ideas which were just drafted in the previous version.
Concerning the lack of references to support a few of their assertions, given the extensive experience of the authors, I bow to the tension between their scientific spirits and accept their anecdotal information as scientific truth. However, I hope that they one day may share with us, earthly scientists, which tend to connect statements with data when writing scientifically, the unpublished knowledge underpinning their confident opinions.